# Self-Supervised Learning of Intertwined Content and Positional Features for Object Detection

**Kang-Jun Liu** [1]  **Masanori Suganuma** [1]  **Takayuki Okatani** [1] [2]

## Abstract

We present a novel self-supervised feature learning method using Vision Transformers (ViT) as the backbone, specifically designed for object detection and instance segmentation. Our approach addresses the challenge of extracting features that capture both class and positional information, which are crucial for these tasks. The method introduces two key components: (1) a positional encoding tied to the cropping process in contrastive learning, which utilizes a novel vector field representation for positional embeddings; and (2) masking and prediction, similar to conventional Masked Image Modeling (MIM), applied in parallel to both content and positional embeddings of image patches. These components enable the effective learning of intertwined content and positional features. We evaluate our method against state-of-the-art approaches, pre-training on ImageNet-1K and fine-tuning on downstream tasks. Our method outperforms the state-of-the-art SSL methods on the COCO object detection benchmark, achieving significant improvements with fewer pre-training epochs. These results suggest that better integration of positional information into self-supervised learning can improve performance on the dense prediction tasks. Our code is available at https://github.com/KJ-rc/IntertwinedSSL.

## 1. Introduction

In recent years, self-supervised learning (SSL) methods (Chen et al., 2020; He et al., 2020; Caron et al., 2020; Grill et al., 2020; Caron et al., 2021; Zbontar et al., 2021; Ermolov et al., 2021) for image feature extraction have advanced significantly. These approaches enable feature extraction from unlabeled images and, with larger training datasets, have improved performance in various downstream tasks (Deng et al., 2009; Geiger et al., 2012; Lin et al., 2014; Wu et al., 2015; Zhou et al., 2017).

While early methods did not restrict the scope of downstream tasks, targeting a broad range from image classification to dense prediction tasks such as semantic segmentation, recent years have seen a shift toward developing SSL methods tailored to specific downstream tasks. This shift stems from the recognition that the features required for image classification, which depends on global image-level representations, differ significantly from those necessary for dense prediction tasks, where pixel-level or patch-level features play a crucial role.

This study builds on recent research trends, focusing specifically on object detection (OD) and instance segmentation (IS) as downstream tasks. These tasks require the precise identification of individual object instances within an image, which necessitates extracting appropriate features from localized regions such as pixels, patches, or subregions. The core focus of this research is on effectively integrating both the content information of these local regions and their positional information within the image into feature representations, as these are arguably crucial for OD and IS tasks.

Among SSL methods, contrastive learning is an early and foundational approach that continues to be widely adopted. In this method, two random crops are taken from a single image, augmented with random transformations, and the feature representations are trained to be similar in the feature space. This approach can be interpreted as focusing on learning position-invariant features for the entire image. Subsequently, pixel-level contrastive learning was introduced to better suit dense prediction tasks. Besides using crops, these methods pair individual pixels and train their features to be similar. A key challenge in this approach lies in determining which pixels to pair. To address this, various strategies have been proposed, including methods based on proximity in feature space (Li et al., 2022a; Su et al., 2024), methods relying on geometric positional correspondence (Yun et al.,

[1]Graduate School of Information Sciences, Tohoku University, Miyagi, Japan [2]RIKEN Center for AIP, Tokyo, Japan. Correspondence to: Takayuki Okatani <okatani@vision.is.tohoku.ac.jp>.

*Proceedings of the 42$^{nd}$ International Conference on Machine Learning*, Vancouver, Canada. PMLR 267, 2025. Copyright 2025 by the author(s).

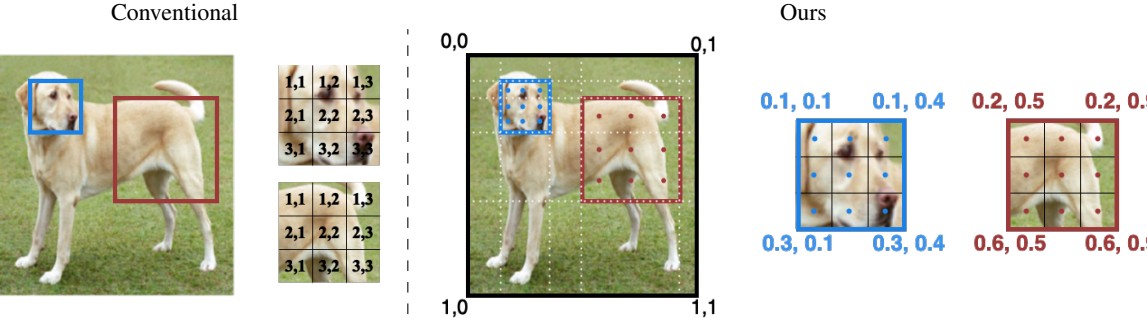

*Figure 1.* **Proposed position encoding.** Conventional methods always apply the same positional encoding to randomly cropped views. In contrast, the proposed method directly applies the positional encoding defined on the original image to the cropped views.

2022; Lebailly & Tuytelaars, 2023), and hybrid approaches that balance these two criteria (Bardes et al., 2022; Lebailly et al., 2024; Stegmüller et al., 2023).

Another line of research focuses on Masked Image Modeling (MIM) (Bao et al., 2022; He et al., 2022; Zhou et al., 2022a), a distinctly different approach. This method, designed to use with Vision Transformers (ViT) (Dosovitskiy et al., 2021), partitions an image into patches and randomly masking a subset of them. The model is trained to reconstruct the masked patches based on the context provided by the unmasked ones. Masked patches as reconstruction targets can be pixels as in MAE (He et al., 2022) or patch-level latent features as in iBOT (Zhou et al., 2022a). MIM is regarded as an effective way to learn feature representations that seamlessly combine both the content information and positional information of each patch.

In this paper, we propose a novel SSL method specifically designed for OD and IS as downstream tasks, building on the contrastive learning framework. These tasks demand both content information and positional information to accurately classify object instances while distinguishing them from one another. Our method focuses on extracting feature representations that seamlessly integrate these two types of information. Although our motivation is similar to recent works such as DropPos (Wang et al., 2023) and LOCA (Caron et al., 2024), our approach introduces two novel components that fundamentally redefine how positional information is utilized during training.

The first is the use of positional encoding tied to the cropping process in contrastive learning; see Figure 1. In conventional SSL (Caron et al., 2021; Chen et al., 2021; Caron et al., 2024), the position encoding is not aligned with the cropping, meaning the same position embeddings are applied whether processing the full image or a cropped sub-image. We propose representing positional encoding as a vector field with the same dimensions as the input image, which is then cropped in the same manner as the image and sampled

on a regular grid, yielding a set of position embeddings of the patches. They are subsequently combined with the content embeddings of their corresponding image patches.

The second component is that, unlike previous MIM (Bao et al., 2022; Peng et al., 2022; He et al., 2022), which apply masking only to the image content embeddings of patches, our method also applies it to position embeddings. In our approach, after the input image is patchified into a set of patches before being fed into the ViT, masking and prediction are performed independently on both their content embeddings and the position embeddings. The underlying expectation is that by predicting the masked positional information from the remaining positional and content information, and vice versa, the model can extract features that intertwine both the image content and positional information. It is important to note that this specialized treatment of positional embeddings is applied only during training, allowing the standard positional embedding method of ViT to be seamlessly employed during inference.

We experimentally compare the proposed method with existing state-of-the-art approaches on the COCO detection dataset (Lin et al., 2014) in the standard setting, i.e., pre-training on ImageNet-1K (Deng et al., 2009) and fine-tuning on COCO. The results show that the proposed method achieves significant performance improvements in the downstream tasks of OD and IS, demonstrating the effectiveness of our approach.

## 2. Preliminaries

**Contrastive self-supervised learning** In SSL, a feature extraction model is trained on a pretext task, and a key is in the design of the task (Doersch et al., 2015; Zhang et al., 2016; Gidaris et al., 2018; Oord et al., 2018; Vincent et al., 2008). Contrastive methods (Oord et al., 2018; Chen et al.,

2020; He et al., 2020)[1] have proven particularly effective, maximizing the similarity of representations from different views of the same image—created through random crops and other diverse image augmentations. These methods ensure that the learned representations are invariant to these augmentations, and they mostly focus on learning image-level representations.

**Self-distillation**  Self-distillation (Caron et al., 2021) is a widely used approach in contrastive SSL. It transfers knowledge from a teacher model to a student model, with both models being updated simultaneously. The student is updated via gradient descent, while the teacher is updated as a momentum copy of the student. Both models, parameterized by $\theta$ and $\theta'$ respectively, share the same network architecture. Given an image $x$ and its two views, $u$ and $v$, which are randomly cropped from $x$ and undergo random augmentations, knowledge is distilled through a cross-entropy loss:

$$\mathcal{L}^{\text{image}} = -P_{\theta'}^{[\text{CLS}]}(v)^{\text{T}} \log P_{\theta}^{[\text{CLS}]}(u), \qquad (1)$$

where $P_{\theta}^{[\text{CLS}]}(\cdot)$ and $P_{\theta'}^{[\text{CLS}]}(\cdot)$ project [CLS] tokens in the VIT's input to a probability distribution over $K$ dimensions. This can also be interpreted as an assignment to $K$ learnable prototypes, with the student learning these assignments from the teacher.

**iBOT (Zhou et al., 2022a) and DINOv2 (Oquab et al., 2024)**  Masked image modeling (MIM) (Bao et al., 2022; He et al., 2022) is an SSL method based on a different principle. Using ViT as the backbone, it divides the input image $u$ into $N$ patches and applies a linear mapping to each, resulting in $N$ vectors $\{u_i\}_{i=1}^{N}$. MIM randomly masks a subset of these vectors and trains a model to predict the masked vectors from the unmasked ones. Specifically, let $\{m_i \in \{0,1\}\}_{i=1}^{N}$ be a random mask sampled according to a ratio $\rho \in [0,1]$. A special token $e_{[\text{MASK}]}$ is introduced, replacing the patch embedding vector $u_i$ with $\hat{u}_i = (1 - m_i) \cdot u_i + m_i \cdot e_{[\text{MASK}]}$ for $i = 1, \ldots, N$. Position embeddings $\{p_i\}_{i=1}^{N}$ are then added to $\hat{u}$ to obtain the integrated embeddings $\{\hat{u}_i + p_i\}_{i=1}^{N}$. iBOT (Zhou et al., 2022a) and DINOv2 (Oquab et al., 2024) implement MIM within a self-distillation framework. The student model $P_{\theta}^{\text{patch}}(\cdot)$ and the teacher model $P_{\theta'}^{\text{patch}}(\cdot)$ project the integrated embeddings into a probability distribution of $K'$ dimensions. The patch-level self-distillation is formulated

---

[1]The term "contrastive methods" in its narrow sense refers to methods that use both positive and negative samples. However, for simplicity in this paper, we also refer to 'non-contrastive methods,' which use only positive pairs, as contrastive.

as the following loss:

$$\mathcal{L}^{\text{patch}} = -\sum_{i=1}^{N} m_i \cdot P_{\theta'}^{\text{patch}}(u_i + p_i)^{\text{T}} \log P_{\theta}^{\text{patch}}(\hat{u}_i + p_i),$$
$$(2)$$

where $P_{\theta}^{\text{patch}}(\cdot)$ and $P_{\theta'}^{\text{patch}}(\cdot)$ share the same network architecture and $\theta$ is updated by gradient descent while $\theta'$ is updated by the exponential moving average of $\theta$. MIM has demonstrated strong performance in dense prediction tasks.

Combining Eq. (1) and Eq. (2), the losses of iBOT (Zhou et al., 2022a) and DINOv2 (Oquab et al., 2024) can be summarized as follows:

$$\mathcal{L}^{\text{DINOv2}} = \underbrace{\mathcal{L}^{\text{image}} + \mathcal{L}^{\text{patch}}}_{\text{iBOT}} + \lambda_{\text{KoLeo}} \mathcal{L}^{\text{KoLeo}}, \quad (3)$$

where $\mathcal{L}^{\text{KoLeo}}$ is defined as in (Sablayrolles et al., 2019) to increase feature diversity. These methods have demonstrated strong performance in both classification and dense prediction tasks.

## 3. Proposed Method

### 3.1. Outline

The objective is to develop a SSL method tailored for object detection (OD) and instance segmentation (IS). Models pre-trained using this method are expected to efficiently extract the necessary features for these downstream tasks and deliver high accuracy with minimal fine-tuning. Following recent studies on SSL methods, we adopt ViT as the backbone for image feature extraction.

As discussed in Section 1, OD and IS require isolating object instances in an image, making it essential to have feature representations that effectively integrate both content and positional information. To address this, the proposed method builds on existing approaches while introducing mechanisms to improve the treatment of positional information in feature learning. Specifically, two key components are introduced. The first is a positional encoding method that aligns with cropping in image-level contrastive (or joint-embedding) techniques. The second is a patch-level masking approach that masks not only image content but also positional information, incorporating both into the prediction target. Each of these components is detailed below.

### 3.2. Position Encoding Linked with Image Cropping

#### 3.2.1. BASIC METHOD

In conventional contrastive methods, two cropped regions from the input image are treated as if they were complete, independent images, and positional embeddings are applied as such (Figure 1). Consequently, the positional and size information of the cropped regions within the original image

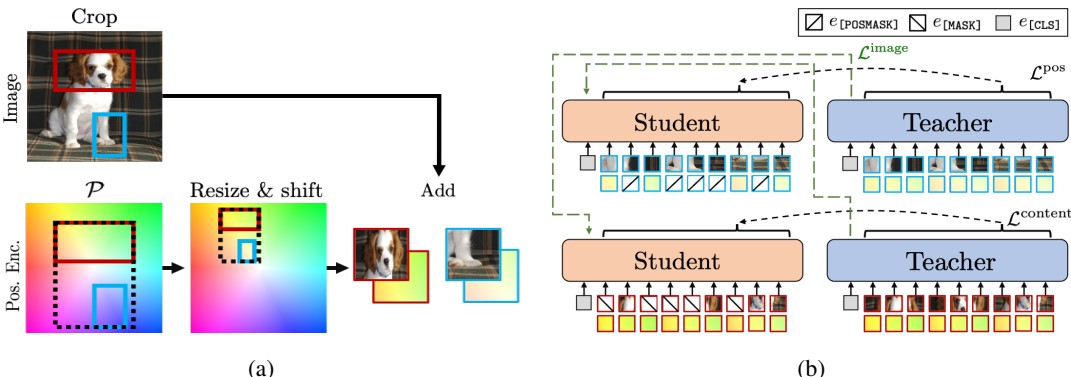

*Figure 2.* **Illustration of the proposed method. (a)** The proposed method defines positional encoding directly on the input image. The position embedding for each view is determined in relation to its crop. Additionally, we introduce a virtual larger image, where the input image is assumed to occupy a random scale and position. Positional embeddings are then calculated within this virtual image. The content embeddings of the image are obtained using standard methods from the original image and are added to the corresponding position embeddings. **(b)** Using these position embeddings, our method performs image-level feature alignment between views ($\mathcal{L}^{\text{image}}$) and independently executes masking and prediction for both position embeddings and content embeddings ($\mathcal{L}^{\text{pos}}, \mathcal{L}^{\text{content}}$).

is completely discarded, which can hinder the accurate identification of object instances. To overcome this limitation, we propose a method that crops the positional information embedded in the original image alongside its RGB content.

The details are as follows. ViT divides the input image $x$ into a fixed number ($N$) of patches, which are then embedded via a linear transformation into a sequence of vectors $\{x_i\}_{i=1}^N$. The position of each patch within the image is encoded by a positional vector $p_i$, which is added to the patch embedding $x_i$ to form $x_i + p_i$. The resulting sequence $\{x_i + p_i\}_{i=1}^N$ is then fed into the input layer of ViT.

Conventional contrastive methods (Chen et al., 2021) extract two cropped views, $u = t_u(x)$ and $v = t_v(x)$, from an input image $x$, and process each view independently in the same way. Specifically, $u$ is divided into patches and embedded as $\{u_i\}_{i=1}^N$, which are combined with the fixed positional encodings $\{p_i\}_{i=1}^N$, resulting in $\{u_i + p_i\}_{i=1}^N$. The same process is applied to $v$, yielding $\{v_i + p_i\}_{i=1}^N$. Thus, while the cropped views $u$ and $v$ can vary in positions and sizes, the positional encodings $p_i$ remain unchanged independently of $t_u$ and $t_v$.

The proposed method introduces an alternative approach to positional encoding. Let $\mathcal{P}$ denote a smooth vector field, with its four corners aligned to those of the input image $x$. While $x$ provides an RGB vector for each pixel, $\mathcal{P}$ encodes the spatial location of each pixel as a vector. Similar to conventional methods, two views, $u = t_u(x)$ and $v = t_v(x)$, are cropped from $x$ and divided into patches, yielding $\{u_i\}_{i=1}^N$ and $\{v_i\}_{i=1}^N$. The same cropping transformations, $t_u$ and $t_v$, are then applied to $\mathcal{P}$, generating cropped vector fields $p_u = t_u(\mathcal{P})$ and $p_v = t_v(\mathcal{P})$.

Subsequently, $p_u$ and $p_v$ are sampled on a regular grid corresponding to the patches, and their values are combined with $u_i$ and $v_i$. The resulting inputs to the Vision Transformer (ViT) are $\{u_i + p_{u,i}\}_{i=1}^N$ and $\{v_i + p_{v,i}\}_{i=1}^N$, where $p_{u,i}$ and $p_{v,i}$ represent the sampled positional vectors for each patch.

In practice, $\mathcal{P}$ is represented as a 2D array of size $w \times h$ consisting of $d$-dimensional vectors, i.e., $\mathcal{P} \in \mathbb{R}^{w \times h \times d}$. The size $w \times h$ are independent of the input image size and are treated as hyperparameters. Due to the arbitrary nature of the crop operations $t_u$ and $t_v$, we interpolate the array $\mathcal{P}$ to obtain $\{p_{u,i}\}_{i=1}^N$ and $\{p_{v,i}\}_{i=1}^N$.

### 3.2.2. POSITION AND SCALE AUGMENTATION

By adopting the above approach, it becomes possible to integrate the position and size of cropped views within the image into the feature representation. However, this method raises two concerns. First, while the approach encodes absolute positional information, the identification of object instances should typically rely only on the relative positional information between the two views; see Figure 1. Learning absolute positions directly and becoming overly dependent on them may lead to unintended consequences.

Second, there is a notable difference in the spatial distribution and size of objects between ImageNet (Deng et al., 2009) images and those used in OD/IS tasks (e.g., COCO). In ImageNet, objects generally occupy a large, centralized region of the image, whereas in COCO, multiple objects of varying sizes appear across different parts of the image. Neglecting this distinction could result in performance issues.

To address these issues, we apply data augmentation over $p_u$ and $p_v$, by randomly shifting and scaling them together

within $\mathcal{P}$; see Figure 2(a). Specifically, let $A$ represent the smallest bounding rectangle that encloses $p_u$ and $p_v$, as shown in Figure 2(a). A coordinate transformation $t_s$, consisting of scaling and translation, is applied to $A$. The scaling factor is $\sqrt{s/|A|}$, where $|A|$ denotes the area of the region. The parameter $s$ is randomly sampled from a Beta distribution, as will be discussed in Section 4.3. The translation (displacement) is randomly sampled from a uniform distribution, ensuring that the new cropped regions remain within the hypothesized input image. With the application of $t_s$, the positional encodings undergo combined transformations, with the cropped fields becoming $p_u \leftarrow t_s(t_u(\mathcal{P}))$ and $p_v \leftarrow t_s(t_v(\mathcal{P}))$.

This position and scale augmentation is expected to reduce excessive reliance on absolute positional information within the image and alleviate biases associated with the object scale distribution in ImageNet.

### 3.3. Masked Position Prediction

The introduction of the above positional encoding method broadens the range of options for designing 'pretext tasks' (i.e., loss functions) during training. In particular, we propose utilizing positional information as a target for Masked Image Modeling (MIM) (Bao et al., 2022; Zhou et al., 2022a; He et al., 2022). Specifically, we extend the approach employed in methods like iBOT and DINOv2—which is traditionally applied exclusively to content embeddings—to include positional embeddings. This extension aims to enhance feature representation learning, with a particular focus on fostering a more effective integration of content and positional information.

The details are as follows. Recall that in our method, the input vector sequence is represented as $\{u_i + p_{u,i}\}_{i=1}^N$, where the positional encoding depends on the cropping of the view $u$. Our masked position prediction works as follows: as in MIM, vectors from the input sequence $\{u_i + p_{u,i}\}_{i=1}^N$ are randomly selected, and the position codes of the selected vectors are masked. Specifically, if $u_i + p_{u,i}$ is selected, it is modified as $u_i + e_{[\text{POSMASK}]}$ using a newly introduced special token $e_{[\text{POSMASK}]}$. The resulting masked sequence is then fed into the ViT. Through preliminary experiments, we observed that when masking position embeddings, selecting patches in a cross-shaped pattern produces better results compared to the box-wise selection used in iBOT and DINOv2 for Masked Image Modeling (MIM) with content embeddings. In those methods, patches are randomly selected in *rectangular boxes*, and all patches within each box are masked until the designated mask ratio $\rho$ is achieved[2]. For further

---

[2]We follow previous studies (Bao et al., 2022; Zhou et al., 2022a; Oquab et al., 2024) in determining $\rho$ for both masked content and positional predictions, where $\rho$ is randomly selected from the range $[0.1, 0.5]$.

details, refer to Section 4.3.

We retain the original content masking and prediction from MIM—specifically, the selected vector $u_i + p_{u,i}$ is modified to $e_{[\text{MASK}]} + p_{u,i}$—but it is performed independently of the position masking and prediction described above; see Figure 2(b) [3] . Thus, masking and prediction are applied symmetrically to both content and position codes. As illustrated in Figure 2(b), this process is applied only to the student side (i.e., only for the view $u$) in the teacher-student framework, similar to the hybrid models introduced after iBOT (Zhou et al., 2022a).

In summary, the position and content masking and prediction are implemented through the following loss functions:

$$\mathcal{L}^{\text{pos}} = -\sum_{i=1}^N P_{\theta'}^{\text{pos}}(u_i + p_{u,i})^T \log P_\theta^{\text{pos}}($$
$$u_i + m^{\text{p}} e_{[\text{POSMASK}]} + (1 - m^{\text{p}})p_{u,i}), \quad (4)$$

$$\mathcal{L}^{\text{content}} = -\sum_{i=1}^N P_{\theta'}^{\text{content}}(u_i + p_{u,i})^T \log P_\theta^{\text{content}}($$
$$m^{\text{c}} e_{[\text{MASK}]} + (1 - m^{\text{c}})u_i + p_{u,i}), \quad (5)$$

where $m_i^{\text{p}} \in \{0,1\}^N$ and $m_i^{\text{c}} \in \{0,1\}^N$ are the sampled masks with a mask ratio $\rho \in [0,1]$.

It is worth noting that some existing methods also incorporate position prediction; however, they predict precise positions using location indicators (Wang et al., 2023; Caron et al., 2024) or at the pixel level (He et al., 2022). In contrast, our method predicts positional information within the feature (embedding) space, representing a fundamentally different approach.

### 3.4. Adaptation to DINOv2

While the proposed method can be adapted to other SSL methods, we focus on integrating it with DINOv2 due to its popularity and performance. The modification to the loss function is straightforward: we add the position masking and prediction loss from Eq. (4) to the original DINOv2 loss as follows:

$$\mathcal{L}^{\text{ours}} = \overbrace{\underbrace{\mathcal{L}^{\text{image}} + \mathcal{L}^{\text{content}}}_{\text{iBOT}} + \lambda_{\text{KoLeo}}\mathcal{L}^{\text{KoLeo}}}^{\text{DINOv2}} + \mathcal{L}^{\text{pos}}, \quad (6)$$

where $\mathcal{L}^{\text{KoLeo}}$ (Sablayrolles et al., 2019) enhances the diversity of image-level representations, and its weight $\lambda_{\text{KoLeo}}$ is set to 1.

---

[3]In our implementation, for each image in a batch, we randomly applied either content or position masking and prediction, each with a 50% probability. These applications are mutually exclusive, meaning both types of masking are never applied to the same image simultaneously.

*Table 1.* **COCO object detection and instance segmentation** and **ADE20K semantic segmentation** We report the results using both ViT-B/16 and ViT-S/16 backbones. The pre-trained weights for all other methods are sourced from their official repositories, except for DINOv2†, which is our reproduction on ImageNet-1K. The highest score is highlighted in bold, while the second-highest is underlined.

| Method | Eff. Ep. [4] | COCO | | ADE20K |
| | | $AP^{Box}$ | $AP^{Mask}$ | mIoU |
| --- | --- | --- | --- | --- |
| *ViT-Small/16* | | | | |
| DINO (2021) | 3200 | 42.0 | 38.0 | 42.9 |
| iBOT (2022a) | 3200 | 43.8 | 39.1 | 44.8 |
| Mugs (2022b) | 3200 | 41.3 | 37.2 | **45.3** |
| CrIBo (2024) | 1600 | 42.6 | 38.3 | 44.9 |
| SelfPatch (2022) | 1050 | 40.4 | 36.7 | 42.5 |
| FLSL | 700 | **45.5** | **40.5** | 41.4 |
| CrOC (2023) | 600 | 40.2 | 36.2 | 43.4 |
| LOCA (2024) | 600 | 40.1 | 36.0 | 44.8 |
| DINOv2† (2024) | 350 | 41.9 | 37.7 | 44.7 |
| **Ours** | 350 | 44.8 | 39.8 | 44.8 |
| *ViT-Base/16* | | | | |
| DINO (2021) | 1600 | 45.5 | 40.8 | 44.7 |
| MAE (2022) | 1600 | 48.1 | 43.2 | 46.2 |
| iBOT (2022a) | 1600 | 47.6 | 42.4 | 47.7 |
| Mugs (2022b) | 1600 | 47.0 | 42.0 | 47.7 |
| DropPos (2023) | 800 | 47.0 | 42.2 | 45.3 |
| CrIBo (2024) | 800 | 45.4 | 40.5 | 45.6 |
| LOCA (2024) | 600 | 48.3 | 43.0 | **48.5** |
| DINOv2† (2024) | 350 | 47.7 | 42.4 | 47.5 |
| **Ours** | 350 | **49.2** | **43.8** | 48.4 |

From an implementation perspective, only minor extensions to the existing DINOv2 code are necessary. Specifically, we compute the additional loss mentioned above. In DINOv2, content masking and prediction were originally applied to 50% of the images in each batch. We now compute the additional position loss for the remaining images. Additionally, the position augmentation described in Section 3.2.2 can be seamlessly integrated into DINOv2's data augmentation pipeline, without requiring any further modifications.

## 4. Experiments

### 4.1. Experimental Configuration

**Pre-training on ImageNet-1K** We pre-train our model on ImageNet-1K (Deng et al., 2009) using the AdamW (Loshchilov, 2017) optimizer for 100 epochs. The multi-crop augmentation is employed following the DINO series (Caron et al., 2021; Zhou et al., 2022a; Oquab et al., 2024), specifically utilizing two global crops and eight local crops. The proposed position encoding method (Section 3.2.2) is

---

[4]Following iBOT (Zhou et al., 2022a), we calculate the effective number of epochs for each method.

applied to both global and local crops, while masked content/position prediction (Section 3.3) is applied only to the global crops. Our training setup is largely based on DINOv2 (Oquab et al., 2024), with several modifications detailed in Appendix B.1. For baseline comparisons, we use a ViT-B backbone, while ablation studies are performed using a ViT-S backbone. The weights for the comparison methods are sourced from their public repositories, except for DINOv2. Since DINOv2 was pre-trained on a much larger dataset of 142M samples (Oquab et al., 2024), we conduct its training on ImageNet-1K, referring to it as DINOv2†.

**Evaluation on COCO and ADE20K** We evaluate the transferability of the features learned by our method on object detection and instance segmentation tasks using the COCO dataset (Lin et al., 2014). We also report performance on ADE20K (Zhou et al., 2017), in line with recent SSL studies (Locatello et al., 2020; Wang et al., 2023). We fine-tune the pre-trained models on COCO and ADE20K as follows. For the COCO dataset, we follow the evaluation methodology from DropPos (Wang et al., 2023), using ViTDet (Li et al., 2022b) as our detection framework while removing window attention and relative position encodings from the backbone. For the ADE20K dataset, we adhere to the evaluation protocol from LOCA (Caron et al., 2024), using the linear decoder approach from Segmenter (Strudel et al., 2021), which utilizes a minimal number of adapter layers. Additional implementation details are provided in Appendix B.

**Compared methods** We compare our method with state-of-the-art SSL approaches. For general-purpose SSL methods, we include DINO (Caron et al., 2021), MAE (He et al., 2022), iBOT (Zhou et al., 2022a), and DINOv2 (Oquab et al., 2024). For SSL methods specifically designed for dense prediction tasks, we evaluate Mugs (Zhou et al., 2022b), DropPos (Wang et al., 2023), CrIBo (Lebailly et al., 2024), SelfPatch (Yun et al., 2022), FLSL (Su et al., 2024), and CrOC (Stegmüller et al., 2023). Explanations for these methods are provided in Section 5. For all methods, we use the implementations and pre-trained weights available from their official repositories. Depending on the availability of configurations, we report results for only one of the two backbones (ViT-B/16 or ViT-S/16) for some methods.

### 4.2. Main Results

**Object detection on COCO** In Table 1, we compare the performance of various SSL methods on the COCO dataset using ViT-S/16 and ViT-B/16 as the backbone.

With one exception, our method outperforms others, including OD/IS-specific SSL methods. The exception is FLSL (Su et al., 2024) with the ViT-S backbone. Although the performance of FLSL is consistent with the results reported

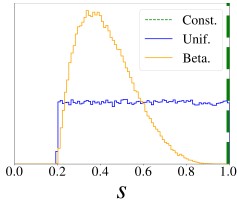

*Figure 3.* **Distributions of scaling factor** $s$

*Table 2.* **Effectiveness of position embedding sampling and masked position prediction** $\mathcal{L}^{\text{pos}}$

| Pos. Sampling | $\mathcal{L}^{\text{pos}}$ | AP$^{\text{Box}}$ | AP$^{\text{Mask}}$ |
|:---:|:---:|:---:|:---:|
| | | 41.3 | . 37.2 |
| ✓ | | 43.2 | 38.6 |
| | ✓ | 43.7 | 39.1 |
| ✓ | ✓ | **44.8** | **39.8** |

*Table 3.* **Effects of hyper-parameters with the proposed position encoding method**

(a) Distributions of scaling factor $s$

| Dist. | AP$^{\text{Box}}$ | AP$^{\text{Mask}}$ |
|:---|:---:|:---:|
| Const. | 43.0 | 38.6 |
| Uniform | 43.7 | 39.0 |
| Beta$(2,5)$ | **44.8** | **39.8** |

(b) Resolutions of $\mathcal{P}$

| Pos. Size | AP$^{\text{Box}}$ | AP$^{\text{Mask}}$ |
|:---|:---:|:---:|
| $19 \times 19$ | 44.4 | 39.7 |
| $50 \times 50$ | **44.8** | **39.8** |

in the original paper, we have identified several issues[5] in its official code[6].

The comparison with other methods is as follows. Our method outperforms all the compared methods in this setting, including OD/IS-specific SSL methods. While Drop-Pos shows some improvement over its base model, DINO, it remains inferior to certain general-purpose methods. We hypothesize that this is due to its high position mask ratio of 94%, which likely hinders the learning of complex visual patterns. With a ViT-B/16 backbone, LOCA performs only on par with the general-purpose MAE, whereas our method surpasses both by +0.9 AP$^{\text{Box}}$ and +0.8 AP$^{\text{Mask}}$. Interestingly, DINOv2$^{\dagger}$ achieves performance comparable to iBOT while requiring only about one-fourth of the effective training epochs, likely benefiting from its broader design exploration.

**Semantic segmentation on ADE20K**  In Table 1, we also report the performance of SSL methods on ADE20K. We observe that the top-performing methods, including iBOT, Mugs, DINOv2$^{\dagger}$, LOCA, and ours, show similar results, with our method not demonstrating a significant improvement in semantic segmentation. This could be because semantic segmentation is primarily a pixel-level classification task. Notably, MAE and DropPos perform worse, likely due to the absence of augmentation invariance in contrastive learning, which may be critical for classification tasks. It is also worth mentioning that the performance of the compared methods is higher than previously reported in LOCA (Caron et al., 2024), likely due to the use of the AdamW optimizer and learning rate scheduler, as recommended by (Lebailly et al., 2024).

### 4.3. Ablation Study

We then examine the design choices in our method by evaluating object detection performance on the COCO dataset. We use a ViT-S backbone here. Starting with the default

settings of our method, we systematically ablate each component or hyperparameter. In all tables, the default configuration is highlighted with a blue background.

**Effectiveness of individual components**  Table 2 shows an ablation study of the two proposed components: the position encoding method and the masked position prediction. Individually, they contribute improvements of +1.9 AP$^{\text{Box}}$ and +2.4 AP$^{\text{Box}}$, respectively. When combined, these modifications further boost performance by at least +1.1 AP$^{\text{Box}}$. These results underscore the effectiveness of each component and indicate their complementary contributions to overall performance.

**Scaling factor** $s$  As described in Section 3.2.2, the proposed positional encoding method simulates feature extraction from small objects within an image by applying random scaling and translation transformations to the crop of the position encoding field. The scaling factor $s$ is sampled from a beta distribution. This approach aligns with the statistical distribution of object sizes in the input images during object detection. To evaluate the impact of different $s$ distributions, we tested three scenarios: $s = 1.0$ (referred to as 'Const.'), a uniform distribution in the range $[0.2, 1.0]$, and Beta$(2, 5)$ as shown in Figure 3. Note that $s = 1.0$ indicates no scaling, meaning the object scale from the original ImageNet image is preserved. The results, shown in Table 3a, indicate that accuracy improves progressively with $s = 1.0$ ('Const'), uniform, and beta distributions, in that order, verifying the proposed sampling method.

**Resolution of position encoding field**  As described in Section 3.2.2, the position-encoding vectors for each patch (e.g., $p_{u,i}$) are obtained by sampling at regular grid points with interpolation from a field $\mathcal{P}$, represented as a tensor of size $w_p \times h_p \times d$. We evaluated the impact of varying the spatial resolution $w_p \times h_p$, a hyperparameter in this representation. The results, shown in Table 3b, indicate that the method is not overly sensitive to resolution as a hyperparameter.

**Position mask sampling strategy**  Through a preliminary study, we found that a cross-wise mask sampling strategy,

---

[5]We attempted to reproduce the SSL pre-training part based on their code. However, we discovered several bugs in the official repo and were unable to reproduce it, as of this writing.

[6]https://github.com/QingSuML

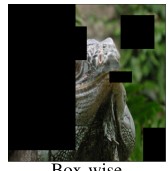 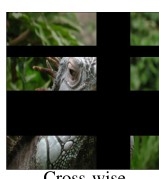

Box-wise         Cross-wise

*Figure 4.* **Position masking: Box-wise vs. Cross-wise**

*Table 4.* **Ablating different position mask settings**

(a) Position masking

| Pos. Mask | $AP^{Box}$ | $AP^{Mask}$ |
|---|---|---|
| Box-wise | 44.4 | 39.7 |
| Cross-wise | **44.8** | **39.8** |

(b) Image ratio in a batch w/ position or content masking

| Content vs. Pos. | $AP^{Box}$ | $AP^{Mask}$ |
|---|---|---|
| 100% w/ Masked Cont. Pred. | 43.4 | 38.9 |
| 100% w/ Masked Pos. Pred. | 21.7 | 20.7 |
| 50% each | **44.8** | **39.8** |

as shown in Figure 4, works effectively for masked position prediction. Table 4a compares this approach with the popular box-wise sampling strategy used for MIM in previous studies, showing that cross-wise sampling performs slightly better. We retain the box-wise sampling scheme for masking content vectors.

**Content masking vs. position masking** Our method applies either content masking and prediction or position masking and prediction to each image in a batch in a mutually exclusive manner. The selection is random, with a default ratio of 50:50. For a sanity check, we also evaluate the configurations of 100:0 and 0:100. The results, shown in Table 4b, indicate that the 50:50 mix achieves the best performance, while applying position masking and prediction alone leads to a significant performance drop. These findings confirm the effectiveness of combining both approaches.

## 5. Related Work

**Dense contrastive learning** Dense contrastive learning (O Pinheiro et al., 2020; Wang et al., 2021; Xie et al., 2021; Bardes et al., 2022) focuses on learning pixel-level representations rather than image-level ones (Oord et al., 2018; Chen et al., 2020; He et al., 2020), with the goal of improving performance on dense prediction downstream tasks such as segmentation and detection. The challenge in pixel-level SSL (Yun et al., 2022; Li et al., 2022a; Lebailly et al., 2024; Bardes et al., 2022; Stegmüller et al., 2023; Lebailly & Tuytelaars, 2023) is positive samples matching problem. Their approaches can be summarized as either similarity-based or position-based, or both. Although (Lebailly et al., 2024; Stegmüller et al., 2023) also track the positions of cropped views in the original image, they still use the conventional position encoding. Therefore, they do not predict relative positions between views and focus only on visual content.

**Self-supervised learning with position prediction** Recently, several studies (Wang et al., 2023; Caron et al., 2024), have aimed to improve performance on dense prediction tasks by incorporating position prediction tasks, which have

long been known as pretext tasks in SSL (Noroozi & Favaro, 2016), into the SSL methods mentioned above. One such method in this line of work is DropPos (Wang et al., 2023), an extension of MAE. In addition to the core principle of MAE, which masks and predicts the content embedding of image patches, DropPos introduces a task where the positional embedding of the patches is 'dropped' and predicted. Specifically, 75% of the tokens $(u_i + p_i)$ are removed, and MAE is applied to the remaining 25%. Among these, 75% (i.e., 18.75% of the total tokens) have their positional embeddings $p_i$ dropped, leaving only $u_i$ as input. The task is to predict the dropped $p_i$ in this configuration. LOCA is an SSL method that processes two views of the input image (a reference view and a smaller query view with overlap) and aims to predict the query view position in the reference view coordinate by a single cross-attention layer. Within the cross-attention layer, it has to identify the overlap parts and predict their positions correctly. In these work, position prediction is formulated as an 1D classification problem:

$$\mathcal{L}^{loc} = \texttt{one\_hot}(i)^T \log P^{loc}(\boldsymbol{x}_i), \qquad (7)$$

where $\texttt{one\_hot}(i)$ is a location indicator, and $P^{loc}(\cdot)$ projects patch tokens (or masked tokens in DropPos) to a probability distribution over $N$ dimensions.

## 6. Summary

We presented a novel self-supervised learning method that learns pre-trained weights optimized for object detection and instance segmentation. The method introduces two key components. The first is a position encoding aligned with cropped views in a contrastive learning setting. This is achieved using a position embedding field, where embedding vectors are sampled on a regular grid corresponding to the geometry of the cropped view in the input image. Combined with the proposed position encoding augmentation, which can be seamlessly integrated into existing SSL data augmentation pipelines, this approach leads to significant improvements on the COCO benchmark compared to DINOv2 (reproduced by us on ImageNet-1K). The second component is the simultaneous masking and prediction of position and content embeddings, further enhancing performance on the COCO benchmark. Our method also performs

comparably to the state-of-the-art LOCA on the ADE20K dataset, where LOCA is specially tuned for this task. We hope this study sheds light on the potential of positional encoding in contrastive learning, an area that remains under-explored in the research community.

## Acknowledgments

This work was partly supported by JST [Moon-shot Research and Development], Grant Number [JPMJMS2032] and JSPS KAKENHI Grant Number 20H05952 and 23H00482.

## Impact Statement

This paper presents work whose goal is to advance the field of Machine Learning. There are many potential societal consequences of our work, none which we feel must be specifically highlighted here.

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

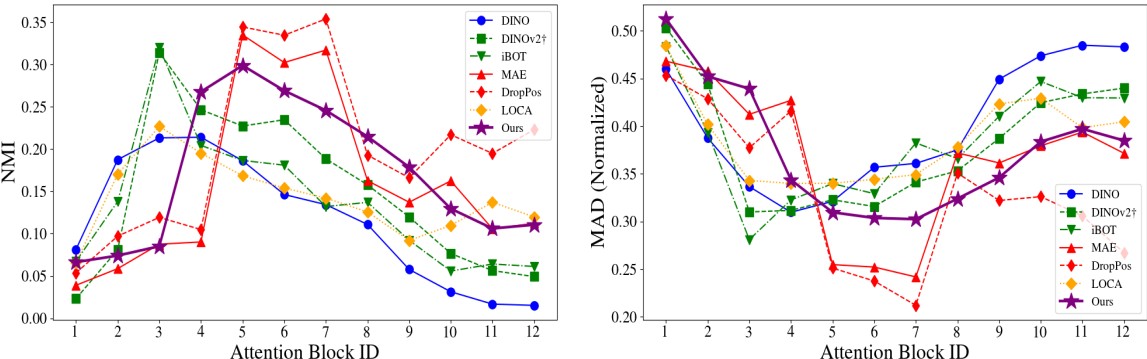

*Figure 5.* **Statistical differences in attention maps across layers of ViT models pre-trained with the compared SSL methods.** Input images are sourced from the COCO dataset. Normalized mutual information (NMI) and mean attention distance (MAD) are used as metrics, following Park et al. (2023). See the text for more details.

## A. Analyses of Attention Maps in Pre-trained Models

Existing SSL methods, including ours, incorporate positional information into feature learning in various ways. Beyond evaluating performance on downstream tasks, we can analyze how attention to patches (tokesn) is spatially distributed within the layers of the ViT, providing insights into the differences between methods. We refer to this spatial distribution as "patch attention" here. In the following, we use pre-trained models from each method, prepared using the same approach as in the previous experiments, and evaluate them. All experiments are conducted using a ViT-B/16 backbone, with input images standardized to $480 \times 480$.

### A.1. Patch attention diversity

Following Park et al. (2023), we first examine the diversity of attention maps and the effective receptive field size. It should be noted that while Park et al.'s analysis uses ImageNet-1K images, we use COCO images as described above. There are two metrics involved (for details, refer to Park et al. (2023)): normalized mutual information (NMI) (Strehl & Ghosh, 2002) and mean average distance (MAD) (Dosovitskiy et al., 2021). NMI is the mutual information between a query token $q$ and a key token $k$, based on the joint distribution $p(q, k) = \pi(k \mid q)p(q)$, where $\pi(k \mid q)$ is the softmax-normalized attention from $q$ to $k$, and $p(q)$ is assumed to be uniformly distributed across the image. Intuitively, this measures the diversity of attention maps. MAD measures the average distance between patch positions within an image, weighted by attention, representing the effective receptive field size in the ViT. The two metrics are computed by averaging over the heads in the attention computation in each layer. The results are shown in Figure 5.

First, it is clear that both metrics vary significantly across methods and layers within the same method. Dividing the methods into four categories—image-level learning (DINO), patch-level learning (MAE and DropPos), hybrid methods (iBOT and DINOv2), and methods incorporating positional learning (LOCA and our method)—the behavior is similar to what was reported in Park et al. (2023). Specifically, higher patch attention diversity is desirable, but it is smaller in image-level methods (DINO) and relatively larger in methods that incorporate patch-level learning. The three methods that showed strong performance on COCO—LOCA, MAE, and our method—exhibit similar behavior in both metrics, particularly in the last three layers.

### A.2. Visualization of patch attention in the final layer

Figure 6 visualizes the attention to all patches (tokens) in the image in the final layer, where a single point (i.e., a patch) in the input image is selected as the query. The attention is averaged across all heads. In the leftmost column of the figure, the position of the query patch is indicated by a red dot in the input image. Across all methods, it appears that patches "close" to the query patch—though in different senses—receive greater attention. However, the interpretation of "closeness" varies across methods. To be specific, MAE can look widely but be biased to textures or colors easily, not enough to identify the instance. Moreover, if the query point is in the background but close to the boundary of a foreground object, attention is

| Input | MAE | DropPos | LOCA | iBOT | Ours |

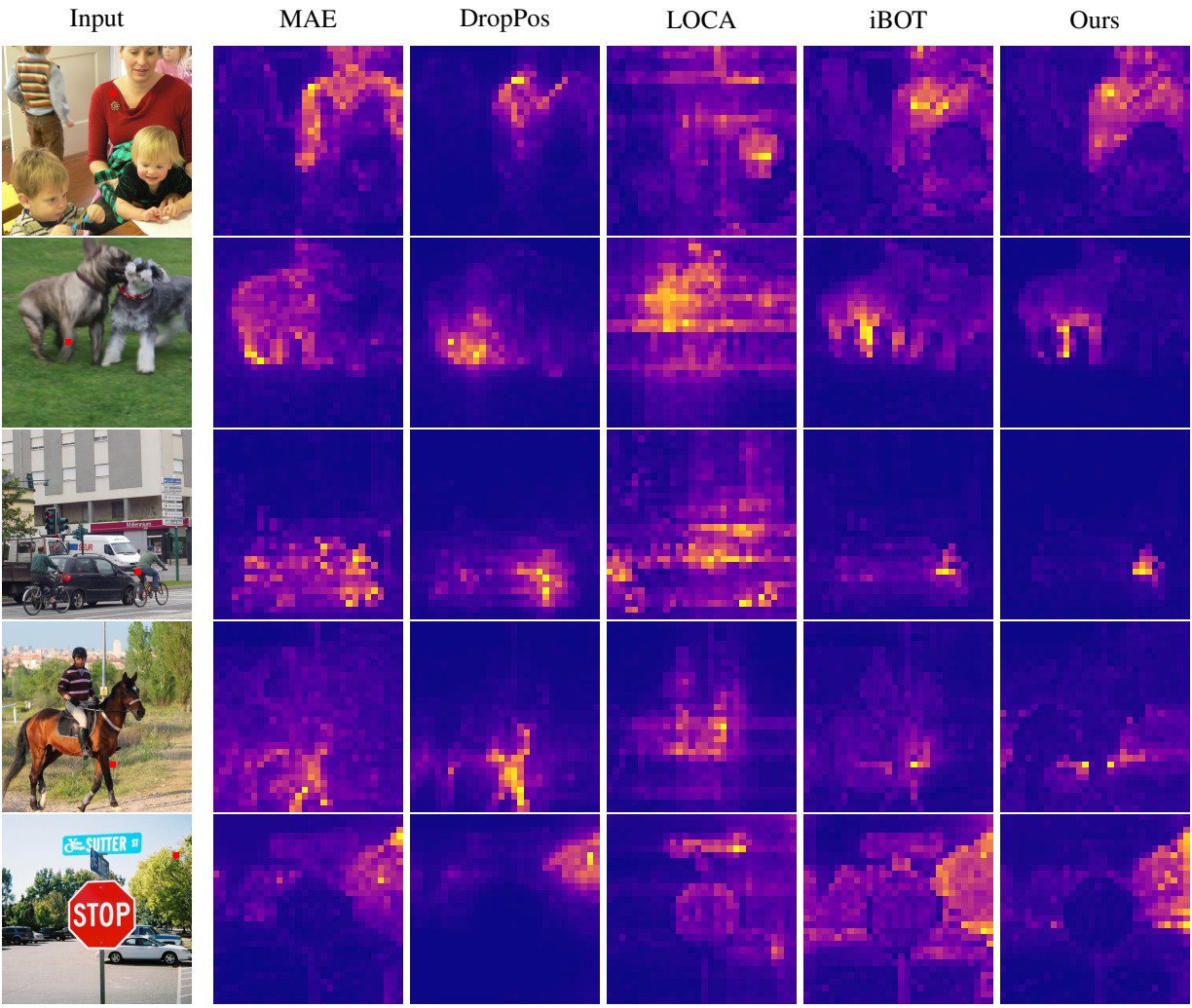

*Figure 6.* **Patch attention maps with sampled reference points as queries.** We visualize the patch attention maps from the final layer, using the sampled reference points (indicated in red) as queries.

focused more on the foreground than the background, as shown in the fourth row. DropPos produces an attention map too locally without recognizing wider patterns. LOCA exhibits cross-shaped artifacts in its attention map. iBOT displays more focused attention than the above methods, but it seems to attend to the entirety of objects in the image without instance discrimination. This is especially evident in the bottom example, where the distinction between trees and the traffic sign is unclear. On the other hand, in the attention maps produced by our method, when the query is on the foreground, the object instance indicated by the query is clearly delineated. When the query is in the background, the background regions can be highlighted more accurately without mixing with the foreground. This behavior suggests that our method is most suitable for object detection and instance segmentation, demonstrating that the proposed approach effectively achieves its goal.

## B. Implementation details

### B.1. Pre-training on ImageNet-1K

We follow the implementation of DINOv2 (Oquab et al., 2024) and adopt some settings from iBOT (Zhou et al., 2022a) due to the significantly smaller scale of training data, i.e., from 142M dataset to ImageNet-1K. We use the same hyperparameters for both the ViT-B and ViT-S backbones, except for the number of GPUs: 8 and 4, respectively. The details are provided in Table 5.

We pre-train ViT-S on a single node with 4 A6000 GPUs and ViT-B on 2 nodes with the same setup. For training time comparison, we compare our method with DINOv2† in our implementation on a single node. Our method requires 1.3× longer training per epoch.

*Table 5.* Implementation details of pre-training on ImageNet-1K

| Config | |
| --- | --- |
| #Epochs | 100 |
| Optimizer | AdamW |
| Base learning rate | 2e-3 |
| Warmup (#epochs) | 10 |
| Layerwise lr decay | False |
| Patch emb. lr decay | False |
| Weight decay (cosine) | 0.04 to 0.4 |
| Drop path rate (linear) | [0, 0.1] |
| Teacher temp. | 0.04 const. |
| Teacher momentum init. | 0.992 |
| Patch mask prob. | [0.1, 0.5] |
| Pos mask prob. | [0.1, 0.5] |
| Patch mask | box-wise |
| Pos. mask | cross-wise |
| Patch/pos mask ratio | 50% vs. 50% |
| Output dim. (all heads) | 65,536 |
| Separate heads | True |
| Norm. last layer | True |
| Dist. of $s$ | Beta(2,5) |
| $s_{min}$, $s_{max}$ ($|\mathcal{P}| = 1$) | 0.2, 1.0 |
| $\mathcal{P}$ | 50×50 |
| Total batch size | 512 |

### B.2. Object detection on COCO with fine-tuning.

Following DropPos (Wang et al., 2023), we adopt ViTDet (Li et al., 2022b) as our object detection framework, fine-tuning the entire model on the COCO object detection benchmark with 22,128 iterations and a total batch size of 64 (12 epochs). We use a 3e-4 learning rate for ViT-B backbone and 1e-4 learning rate for ViT-S backbone. Both learning rate decay at the 19,667-th and 21,306-th iterations by a factor of 10. To preserve the integrity of the pre-trained weights, we remove relative position encodings and window attentions from ViTDet, ensuring that the backbone remains as close as possible to its original pre-trained configuration. We also employ checkpointing and efficient attention kernels (Lefaudeux et al., 2022) to optimize GPU memory usage.

### B.3. Semantic segmentation on ADE20K with fine-tuning

For the ADE20K dataset, we follow LOCA (Caron et al., 2024), adopting the linear decoder protocol in Segmenter (Strudel et al., 2021) and training for 127 epochs with a batch size of 16 (resulting in a total of 160k iterations). We consider the optimizer and learning rate settings from Lebailly et al. (2024), and employ the AdamW optimizer, and sweep the weight decay (wd) across {1e-2, 1e-4} and the learning rate (lr) across {8e-5, 3e-5, 1e-5, 8e-6} with a min_lr of 0.1× lr. We report results in single scale, averagd over 2 runs. This largely improves the performances reported in LOCA. The codebase is based on Segmenter (Strudel et al., 2021) and built using MMSegmentation (Contributors, 2020).

## C. Qualitative studies of position masking: Box-wise vs. Cross-wise

We visualize that vertical line artifacts occur with the box-wise strategy but are absent with the cross-wise strategy; see Figure 7. Furthermore, the superiority of the cross-wise strategy is quantitatively validated in Table 4a.

|  | Input | Box-wise | Cross-wise |

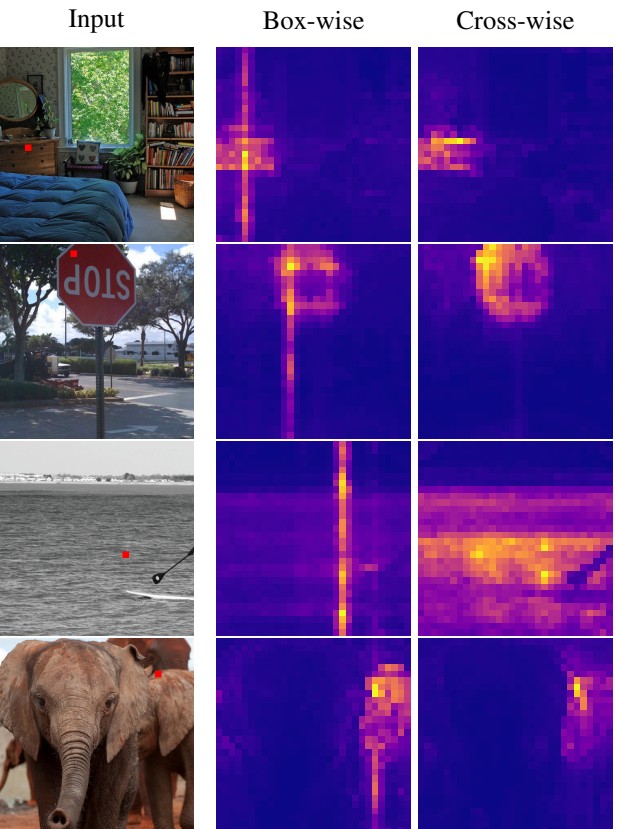

*Figure 7.* **Artifact comparison in attention maps: Box-wise vs. Cross-wise.** Vertical line artifacts occur with the box-wise strategy but are absent with the cross-wise strategy

## D. Comparison with Official DINOv2 and DINOv2-reg Backbones

We augment the results in the main paper with evaluations using pre-trained weights from DINOv2 (Oquab et al., 2024) and DINOv2-reg (Darcet et al., 2024), which were pre-trained on the large-scale LVD-142M dataset. We adopt them in our fine-tuning experiments on COCO and ADE20K, as shown in Table 6. It is important to note that their ViT-S/B models are distilled from a larger pre-trained model, ViT-g, and refined with a larger resolution fine-tuning. Since they use a patch size of 14×14, we interpolate the patchifier kernel from 14×14 to 16×16 before fine-tuning to ensure a fair comparison.

DINOv2-reg and DINOv2 achieve $52.2$ $AP^{Box}$ on the ViT-B/16 backbone and a $47.4$ $AP^{Box}$ on the ViT-S/16 backbone respectively, which are at least $+2$ $AP^{Box}$ higher than the other methods. Interestingly, we find that DINOv2-reg significantly outperforms DINOv2 with ViT-B but not with ViT-S.

*Table 6.* **Augmented results with official DINOv2/-reg (Oquab et al., 2024; Darcet et al., 2024) backbones.**

(a) **ViT-B/16** backbone.

| Method | Eff. Ep. | COCO | | ADE20K |
| | | $AP^{Box}$ | $AP^{Mask}$ | mIoU |
|---|---|---|---|---|
| DINO | 1600 | 45.5 | 40.8 | 44.7 |
| MAE | 1600 | 48.1 | 43.2 | 46.2 |
| iBOT | 1600 | 47.6 | 42.4 | 47.7 |
| Mugs | 1600 | 47.0 | 42.0 | 47.7 |
| DropPos | 800 | 47.0 | 42.2 | 45.3 |
| CrIBo | 800 | 45.4 | 40.5 | 45.6 |
| LOCA | 600 | 48.3 | 43.0 | 48.5 |
| DINOv2† | 350 | 47.7 | 42.4 | 47.5 |
| Ours | 350 | 49.2 | 43.8 | 48.4 |
| *On LVD-142M* | | | | |
| DINOv2 | - | 51.1 | 45.3 | 52.5 |
| DINOv2-reg | - | **52.2** | **46.3** | **54.3** |

(b) **ViT-S/16** backbone.

| Method | Eff. Ep. | COCO | | ADE20K |
| | | $AP^{Box}$ | $AP^{Mask}$ | mIoU |
|---|---|---|---|---|
| DINO | 3200 | 42.0 | 38.0 | 42.9 |
| iBOT | 3200 | 43.8 | 39.1 | 44.8 |
| Mugs | 3200 | 41.3 | 37.2 | 45.3 |
| CrIBo | 1600 | 42.6 | 38.3 | 44.9 |
| SelfPatch | 1050 | 40.4 | 36.7 | 42.5 |
| CrOC | 600 | 40.2 | 36.2 | 43.4 |
| LOCA | 600 | 40.1 | 36.0 | 44.8 |
| DINOv2† | 350 | 41.9 | 37.7 | 44.7 |
| Ours | 350 | 44.8 | 39.8 | 44.8 |
| *On LVD-142M* | | | | |
| DINOv2 | - | **47.4** | **42.2** | **49.7** |
| DINOv2-reg | - | 46.5 | 41.5 | **49.7** |

