# OpenReview forum: "Self-Supervised Learning of Intertwined Content and Positional Features for Object Detection"
_ICML.cc/2025/Conference — ICML 2025 poster_

### Official Review · Reviewer_GJjf · 2025-03-07

**Overall Recommendation:** 3

**Summary:**

Post-rebuttal

After reading the comments by reviewers v6Kv and mPP8, I agree that contrastive learning for dense prediction can offer advantages over autoencoder-based methods, such as more efficient training without heavy fine-tuning. I also agree that the proposed method introduces some technical novelty beyond DropPos. For these reasons, I have updated my rating to weak accept.

---

This paper enhances dense representation learning from contrastive learning by 1) using position encoding as a relative location within the entire image instead of renormalizing it in each cropped image, and 2) reconstructing position information to preserve spatial structure. As a result, the proposed method outperforms or is on par with previous dense contrastive learning methods such as MuGS, FLSL, and LOCA.

**Claims And Evidence:**

I'll combine this section with "Methods and Evaluation Criteria."

**Essential References Not Discussed:**

There is a vast amount of work on (dense) contrastive learning, so I understand that the paper cannot cover all of them. However, it should at least cite [1], as it is highly relevant to the position prediction proposed in this paper.

**Experimental Designs Or Analyses:**

Major concerns were discussed above. Here are some minor comments:

- DropPos is from NeurIPS 2023, not 2024, meaning it was released more than a year ago.

- The DINOv2 results are misleading since the model was designed for larger-scale datasets. However, this paper reimplemented it on IN-1k, leading to significantly lower performance compared to other methods. It would be better to report both the original DINOv2 and the reimplemented version while clearly stating that the original DINOv2 performs better due to its larger training dataset.

**Methods And Evaluation Criteria:**

I have several concerns regarding the motivation, method, and evaluation.

A. Why use contrastive learning for dense representation? Many works since MIM/MAE have shown that pixel-level objectives learn better dense representations than contrastive learning. For instance, Table 4 of the MAE (CVPR 2022) paper shows that ViT-B trained with MAE on IN-1k achieves 50.3 and 44.9 box and mask AP, respectively, outperforming this paper’s 49.2 and 43.8, as reported in Table 1. What advantage does this method have over MAE and its numerous follow-ups, including those combining MAE with contrastive learning?

B. How does this method perform on tasks beyond object detection and segmentation? It is well known that dense contrastive learning involves a trade-off between high-level tasks (e.g., classification) and low-level tasks (e.g., detection). I suspect this method may degrade classification performance. Could you report linear probing accuracy on IN-1k to verify this?

C. The technical novelty is limited. The idea of position-aware cropping has been explored since DenseCL, with various methods adopting their own variants, including LOCA, as mentioned in this paper. Using global positional embeddings may serve as an additional trick for ViT-based contrastive learning, potentially encoding the scale of cropped images. However, its precise effect is not well justified. For instance, it introduces a distribution shift in positional encodings between training and testing, which could harm image-level prediction, as full [0-1,0-1] encodings are always used at test time, potentially reducing scale invariance in classification.

D. Position reconstruction has already been widely explored. This includes methods such as DropPos, as mentioned in the paper. Additionally, MP3 [1] is an example of a missing related work.

E. Why not explore contrastive learning on multi-object images? IN-1k is mostly object-centric, but region-aware contrastive learning would be more impactful for complex, multi-object images. LAION would be a practical example, but early academic papers [2,3] have demonstrated this on smaller datasets like COCO. Notably, region-aware contrastive learning can also improve image classification by preventing representation collapse, benefiting not just detection and segmentation.

[1] Position Prediction as an Effective Pretraining Strategy. ICML 2022.\
[2] Demystifying Contrastive Self-Supervised Learning: Invariances, Augmentations, and Dataset Biases. NeurIPS 2020.\
[3] Object-aware Contrastive Learning for Debiased Scene Representation. NeurIPS 2021.

**Other Comments Or Suggestions:**

N/A

**Other Strengths And Weaknesses:**

N/A

**Questions For Authors:**

All my concerns have been discussed above.

**Relation To Broader Scientific Literature:**

N/A

**Theoretical Claims:**

N/A

---

> ### Author Rebuttal · Authors · 2025-04-01
>
> **Q1.  Why use contrastive learning for dense representation? ...**
>
> The importance of this study can be explained from practical and exploratory perspectives.
> Practical Perspective: Although the CVPR2022 MAE results demonstrate high object detection performance, they incur extremely high computational costs during both pre-training and fine-tuning (specifically, 1600 epochs for pre-training and 100 epochs for object detection fine-tuning). In contrast, in our experiments (and those of related studies), pre-training is conducted in 350-700 epochs and fine-tuning requires only 12 epochs; under these conditions, MAE yields inferior results. In other words, our approach achieves satisfactory performance while significantly reducing computational cost.
> Exploratory Perspective: Our results offer new insights into the challenge of determining what information should be extracted from images for dense prediction tasks. They also show that even within the framework of contrastive learning, these tasks are learnable, and there may be further room for improvement. We expect that our work will serve as a stepping stone for future discoveries.
>
> **Q2. How does this method perform on tasks beyond object detection and segmentation? ...**
>
> As you correctly pointed out, our method, like existing approaches, also shows relatively low performance on classification tasks. The linear probing accuracy on ImageNet-1K is 76.8% (evaluated using ViT-B/16 with DINOv2’s linear evaluation protocol). However, we would like to emphasize that we do not consider this a disadvantage. Regardless of the relative "level" of the tasks (if anything, we consider object detection to be a higher-level task, as it involves both classification and geometric estimation), the two tasks are inherently different to some extent—for instance, in the way positional information is utilized. Therefore, we view this trade-off as a natural consequence.
>
> **Q3. & Q4. The technical novelty is limited.  ...;  Position reconstruction has already been widely explored. ...**
>
> Thank you for your comment. We respectfully contend that our proposed method is genuinely novel and should not be regarded as merely a variant of existing approaches. While it shares the common objective of embedding positional information into feature representations, its approach differs significantly from methods such as DropPos, MP3, and LOCA.
>
> Moreover, because simple position prediction tasks tend to overfit, DropPos uses a high masking rate (94%) and MP3 removes all positional tokens. Our method avoids these measures, demonstrating both its effectiveness and novelty. Although there may be some adverse effects on image-level prediction (classification), this is not our primary target; as reported in the table below for few-shot classification following the settings in MSN[1], the performance drop in classification is smaller than that of other methods with similar objectives.
>
> Finally, regarding the concern about the justification of our method’s effectiveness, our results on dense prediction tasks—supported by comprehensive evaluations (in Table 1), ablation studies (in Table 2), and attention analyses (Figure 6)—clearly show that our approach outperforms conventional methods. We are prepared to conduct additional experiments if necessary and appreciate the comment regarding MP3, which we plan to address in the revised manuscript.
>
> | Method | 1 Img/class | 2 Img/class | 5 Img/class | 1%   |
> |---|--|---|----|------|
> | MAE    | 6.0±0.2  | 9.6±0.3 |  17.5±0.2 | 28.3  |
> | LOCA   | 31.9±0.2 | 40.3±0.4 | 49.7±0.1 | 56.9 |
> | Ours   | 39.2±0.3 | 48.3±0.4 | 56.9±0.1 | 63.0 |
>
> Note: Unlike MSN, we report linear probing results for MAE without partial fine-tuning.
>
> [1] Assran, Mahmoud, et al. "Masked siamese networks for label-efficient learning." ECCV, 2022. (KJ-note. Adding the performance of iBOT for few-shot.)
>
> **Q5. Why not explore contrastive learning on multi-object images?...**
>
> In our study, we primarily focused on learning feature representations that benefit dense prediction tasks. However, as you noted, SSL on multi-object/scene images has also garnered significant interest in the community. To explore this direction, we conducted similar experiments using the Object365 dataset—a dataset of a manageable size—instead of ImageNet-1K. Without any hyperparameter tuning, we achieved 48.7 mAP on the object detection task, which is nearly equivalent to the 49.1 mAP obtained with ImageNet-1K. This demonstrates that our approach performs comparably on more challenging, multi-object images. We plan to further evaluate our method across a wider range of tasks beyond dense prediction in future work.
>
> **To minor comments.**
>
> We will revise the reference for DropPos to correctly cite NeurIPS 2023. We will also note the DINOv2 results by reporting the performance from its original model trained on larger datasets and highlighting the difference in training conditions and expected performance.

---

### Official Review · Reviewer_dcuT · 2025-03-13

**Overall Recommendation:** 3

**Summary:**

The paper proposes a novel self-supervised learning (SSL) framework tailored specifically for object detection (OD) and instance segmentation (IS). The key idea is to integrate positional information more effectively by introducing a learnable positional encoding field that is aligned with the image cropping process. In addition, the method employs a dual masking strategy where both the content and positional embeddings of image patches are masked and then predicted. The authors report improvements on COCO for both detection ($AP_{Box}$) and instance segmentation ($AP_{Mask}$), and they also evaluate on ADE20K for semantic segmentation.

## Update after rebuttal
I appreciate the authors’ efforts in preparing the rebuttal and addressing the major concerns I raised. While I had suggested experimenting with a larger backbone to facilitate a more direct comparison with a state-of-the-art method, I recognize that this concern is relatively minor. Given the authors’ thorough clarifications and convincing responses, I have decided to adjust my overall assessment to Weak Accept.

**Claims And Evidence:**

Claims:
- By aligning the positional encoding with the cropping process, the method preserves crucial spatial cues that are lost in conventional approaches.
- Simultaneously masking and predicting both content and positional embeddings leads to improved feature representations, especially beneficial for OD and IS tasks.
- The proposed approach achieves competitive or superior performance on COCO compared to existing methods.

Evidence:
- Experimental results on COCO indicate improvements in AP_Box and AP_Mask compared to several baselines.
- Ablation studies demonstrate that both the cropping-aligned positional encoding and the dual masking strategy contribute to performance gains.

However, the direct comparison with LOCA - which also focuses on incorporating positional information - is limited. Further experiments (e.g., using a ViT-Large/16 backbone or few-shot settings) would strengthen the evidence for the method’s claimed advantages.

**Essential References Not Discussed:**

The paper cites most of the relevant works in SSL and dense prediction tasks.

**Experimental Designs Or Analyses:**

The experimental design includes:
- Pre-training on ImageNet-1K using a ViT-based architecture.
- Evaluation on COCO for OD/IS and ADE20K for semantic segmentation.
- Detailed ablation studies analyzing the effects of different hyperparameters and design choices (e.g., mask sampling strategies, resolution of the positional field).

Concerns:
- The current experiments, while solid, could benefit from closer alignment with the settings used in LOCA. For instance, experiments with a stronger backbone like ViT-Large/16 and evaluations under few-shot scenarios would provide more comprehensive evidence for the method’s effectiveness on OD/IS.

**Methods And Evaluation Criteria:**

The proposed method is well-motivated for OD and IS because these tasks require precise localization and instance-level discrimination, which rely heavily on accurate positional information. The evaluation uses standard benchmarks - COCO for OD/IS and ADE20K for semantic segmentation - and includes extensive ablation studies to assess different design choices.

Concerns:
- While semantic segmentation is evaluated, the paper emphasizes OD and IS as the primary target tasks; the authors should clarify why they separate these tasks given that semantic segmentation also requires fine positional accuracy.
- To better emphasize OD and IS, additional datasets specific to each task could be included.

**Other Comments Or Suggestions:**

N/A

**Other Strengths And Weaknesses:**

Strengths:
- Provides comprehensive ablation studies that detail the contribution of each component.
- Experimental results on COCO demonstrate improvements in both detection and instance segmentation metrics.

Weaknesses:
- Figure 2 could benefit from additional annotations or breakdowns to clarify the process and key components of the proposed method.
- The lack of provided code hinders reproducibility and a deeper understanding of the implementation details.

**Questions For Authors:**

1. Have you experimented with stronger backbones such as ViT-Large/16? It would be valuable to see experiments that use a higher-capacity model to evaluate how your approach scales. Positive results in this setting would not only solidify your claims about the method’s effectiveness for dense prediction tasks but also offer a more direct comparison to competitors like LOCA, which have explored similar configurations.
2. Have you explored the performance of your method under limited-dataset conditions for OD and IS? By evaluating your approach in few-shot settings - where only a limited amount of training data is available - you could further substantiate the claim that your method learns robust representations that are effective even with minimal supervision. For example, if your method can achieve competitive performance with a few-shot setup similar to that reported for LOCA, it would significantly reinforce your argument regarding its suitability for real-world applications.
3. Could you elaborate on why you chose to focus specifically on object detection (OD) and instance segmentation (IS) rather than addressing semantic segmentation (SS) in a unified manner? The current motivation appears to emphasize locating objects rather than handling instance-level prediction uniquely. This raises concerns about whether the method is truly specialized for OD and IS as opposed to SS. A more detailed explanation of the distinct challenges and requirements of OD and IS would help justify your targeted approach and clarify the specific advantages your method offers for these tasks.
4. Have you considered evaluating your method on additional datasets dedicated specifically to OD or IS? Although COCO is a widely recognized benchmark and provides a strong basis for evaluation, incorporating experiments on another dataset would help demonstrate that the improvements observed are not unique to COCO.

**Relation To Broader Scientific Literature:**

The paper builds on a rich body of SSL work including methods like DINO, iBOT, and DINOv2, and it specifically addresses limitations in previous approaches that attempt to integrate positional information. Although the idea of leveraging positional cues is not entirely new, the paper’s approach of using a cropped positional field and dual masking is presented as an incremental yet meaningful improvement. However, since LOCA is the most direct competitor in this area, a more rigorous experimental comparison is needed to highlight the distinct advantages of the proposed method.

**Theoretical Claims:**

There are no formal theoretical proofs provided in the paper. The contribution is primarily empirical, with the rationale for the positional encoding strategy based on intuitive arguments and validated through experiments.

---

> ### Author Rebuttal · Authors · 2025-04-01
>
> **Q1. Have you experimented with stronger backbones such as ViT-Large/16?...**
>
> Due to insufficient computational resources, we have not been able to conduct such large-scale experiments comprehensively; however, preliminary results confirm the scalability of our approach. Nonetheless, we believe that the performance of our proposed method has been sufficiently demonstrated by the results reported in the paper.
>
> **Q2. Have you explored the performance of your method under limited-dataset conditions for OD and IS?...**
>
> We thank the reviewer for the suggestion. To further evaluate our method's performance under limited training data conditions, we conducted fine-tuning experiments on OD and IS. The results are shown in the table below. Here, the labels 1/32, 1/8, etc. indicate that only 1/32, 1/8, etc. of the original dataset (randomly selected) was used for training, following the experimental settings of LOCA. These results confirm the superiority of our method even under limited-dataset conditions.
>
> | Method | 1/32 AP$^\text{Box}$ | 1/32 AP$^\text{Mask}$ | 1/8 AP$^\text{Box}$ | 1/8 AP$^\text{Mask}$ | 1/2 AP$^\text{Box}$ | 1/2 AP$^\text{Mask}$ | 1 AP$^\text{Box}$ | 1 AP$^\text{Mask}$ |
> |--------|----------------------|-----------------------|---------------------|----------------------|---------------------|----------------------|-------------------|--------------------|
> | MAE    | 31.0                 | 28.8                  | 33.1                | 31.1                 | 42.6                | 39.0                 | 48.1              | 43.2               |
> | LOCA   | 30.4                 | 28.2                  | 33.0                | 30.6                 | 42.3                | 38.5                 | 48.3              | 43.0               |
> | Ours   | 33.0                 | 30.8                  | 32.8                | 30.6                 | 43.3                | 38.9                 | 49.2              | 43.8               |
>
> **Q3. Could you elaborate on why you chose to focus specifically on OD/IS rather than addressing SS in a unified manner?**
>
> As you correctly noted, both OD/IS and SS rely on pixel-level class and positional information, yet their requirements for positional precision differ. In OD/IS, it is crucial to accurately distinguish individual instances—even when they are adjacent or overlapping within the same class. Consequently, OD/IS tasks demand the ability to extract and integrate detailed positional information with class cues—precisely the capability our proposed method is designed to achieve. In contrast, SS primarily benefits from positional information that ensures the continuity and consistency of regions corresponding to the same class. Thus, the fine-grained, class-integrated positional cues essential for OD/IS are not only unnecessary for SS but, given the constraints of representational capacity, may even be counterproductive.
>
> **Q4. Have you considered evaluating your method on additional datasets dedicated specifically to OD or IS?**
>
> We conducted experiments on LVIS (with ViT-B/16, and 12 fine-tuning epochs) and observed a similar tendency to that on COCO, as shown in the table below.
>
> | Method | AP$^\text{Box}$ | AP$^\text{Mask}$ |
> |--------|-----------------|------------------|
> | MAE    | 30.1            | 28.7             |
> | LOCA   | 29.6            | 29.6             |
> | Ours   | 30.6            | 29.6             |
>
>
> **To other weaknesses**
>
> We appreciate the feedback. We will revise Figure 2 to include clearer annotations and breakdowns to better highlight the key components of our method. Additionally, we plan to release the full codebase to ensure reproducibility and facilitate further exploration by the community.

---

### Official Review · Reviewer_mPP8 · 2025-03-14

**Overall Recommendation:** 4

**Summary:**

The paper proposes and investigates a novel positional encoding method and an extension to DINOv2 loss that incorporates positional masking for better SSL pretraining of Object Detection and Instance Segmentation Vision Transformer backbones. The method achieved competitive performance on COCO and ADE20K. Ablation studies show the effectiveness of the proposed components, positional sampling, and positional masking. The study also investigates a statistical ground for its scale distribution for the positional sampling part and empirical support for its positional masking strategy and mixed content-position prediction.

## update after rebuttal

This is an interesting work to me. I like that they incorporate relative positional information in the pretraining by using their proposed positional encoding. The experiments displays some questionable points, as noted by the reviewers, but are overall promising. So it come to whether we should take reasonably projected results or ask for apple-to-apple comparisons, and I'll leave it to you.
The comparison between masking strategies is not significant. However, I probably should not just focus on the numbers. This work displays a complete path from identifying the cause of the attention artifacts to solving it by designing a masking strategy. The question of how the attention artifacts are less relevant to the final outcome is not a burning question that hinders acceptance.
I am not concerned of the motivation. The question of why contrasive SSL for segmentation is reasonable and worth asking, but not very constructive if taken very seriously. Should it be constructive, then why SSL at all, we can always finetune.
Therefore I deem this work to be worth displaying to the community for its progress in processing position information with SSL for segmentation.

**Claims And Evidence:**

This work claims that feature representation integrating content and positional information is important for OD and IS tasks. By introducing new cropping-aware positional encoding and position-masking during training, the ViT models studied can learn such features better. The claim is well-supported by comparative studies to other recent works in image SSL and ablation studies of the proposed components.

**Essential References Not Discussed:**

No

**Experimental Designs Or Analyses:**

Yes. The work follows the evaluation protocol of previous works. For OD on COCO, they use the protocol from DropPos and framework from ViTDet while removing windowed attention and RoPE. IS on ADE20K uses the protocol from LOCA and Segmenter’s linear decoder for minimal adaptation. Methods are compared under the same pipeline, and the results are strong under standard metrics: average precision for COCO and mean intersection over union for ADE20K.

**Methods And Evaluation Criteria:**

Yes. The paper proposes novel positional embedding and modified DINOv2 loss, adding positional prediction to improve self-supervised learning for Object Detection and Instance Segmentation. The datasets of choice are COCO and ADE20K, which are commonly tested datasets for OD and IS, respectively. The evaluation metrics are box/mask average precision and mean intersection over union for COCO and ADE20K, respectively. It makes sense.

**Other Comments Or Suggestions:**

Though this work is positioned for improving IS and OD performance, displaying ImageNet classification results could make this work more valuable to readers. A line plot of different ratios between position masking and content masking v.s. performance is another thing that interests me, potentially others.
L362 - Table Table 3a: delete one “Table”
Figure 3 is dangling with no reference to it in the main text.
L643 - Inconsistent usage of Figure and Fig.

**Other Strengths And Weaknesses:**

One weakness of this work is the unclear cause of the stronger vertical artifact, compared to horizontal, in the attention map when box-based masks are used.

**Questions For Authors:**

1. Is the choice of scale factor distribution based on intuition or mathematical grounds?
2. Why is horizontal attention artifact absent or weaker than vertical ones? Does it imply that the vertical position is harder to predict for images in the pretraining dataset?
3. Does an imbalanced content vs position image ratio improve the results?

**Relation To Broader Scientific Literature:**

This work extends traditional discrete positional encoding to a normalized vector space and uses it to incorporate relative positional/scale information of image patches in the embedding process. The work demonstrated stronger performance than traditional sin/cos positional encoding on OD and IS tasks.

The authors discussed that the position inference target connects to DropPos and LOCA. This work combines it with pixel inference and proposed novel positional encoding to achieve better performance. Also, the contrasive learning target is based on iBOT and DINO, which are strong image self-supervised learning methods. This work expands SSL for OD and IS by adding augmentation in positional encoding and combining positional inference and pixel inference.

**Theoretical Claims:**

The paper is highly empirical with minimal theoretical claims. However, in Section 4.3, there is a claim that relative object scale in object detection follows beta distribution with no further explanation. The ablation study shows a connection between these two by displaying a better result. Should the writer want to make an empirical observation, it is better to state it clearly.

---

> ### Author Rebuttal · Authors · 2025-04-01
>
> **W1. One weakness of this work is the unclear cause of the stronger vertical artifact, compared to horizontal, in the attention map when box-based masks are used.**
>
> Two potential explanations may account for the artifact being more pronounced vertically than horizontally: the statistical bias inherent in the image content and the effect of the horizontal flip (hflip) applied during training. To investigate these factors, we conducted two experiments. Note that although typical contrastive learning applies hflip independently to each crop, our method applies it uniformly to the entire input image to maintain consistent spatial relationships across views.
>
> First, to assess the impact of image content, we rotated all training images 90° counterclockwise and then trained the model under otherwise identical conditions (with hflip applied as usual). As a result, the artifact also rotated 90°, appearing horizontally rather than vertically when we fed test images in their original orientation.
>
> Next, we augmented the data by randomly applying a vertical flip (vflip) at a 1:1 ratio alongside hflip. We expected this to mitigate or randomize any directional artifacts; however, the horizontal artifact persisted.
> These findings suggest that the predominance of the vertical artifact is primarily driven by the intrinsic characteristics of the image content. It appears that ImageNet—and natural images in general—exhibit different statistical properties along the vertical and horizontal axes. Further investigation into this phenomenon is warranted for future work.
>
> We provide a figure at the following URL illustrating the effect of each transformation (Hflip, Rot90, hflip/vflip) on the resulting attention maps for reference.
>
> https://drive.google.com/file/d/1uezT10nsOwK6KtWqvBSN-so6B6DdiRUe/view?usp=drive_link
>
> **Q1. Though this work is positioned for improving IS and OD performance, displaying ImageNet classification results could make this work more valuable to readers.**
>
> Thank you for your suggestion. We will include the ImageNet classification results in the revised manuscript. In summary, our method's performance is slightly lower than that of DINOv2* (our reproduction on ImageNet-1k), yet it outperforms existing SSL methods designed for dense prediction. We report the linear probing performance below, following the protocol used in DINOv2.
>
> | IM1K; Linear Probing | ViT-B/16 | ViT-S/16 |
> |----------------------|----------|----------|
> | DINOv2*              | 78.2     | 74.4     |
> | Ours                 | 76.8     | 71.6     |
>
>
> **Q2. Is the choice of scale factor distribution based on intuition or on mathematical grounds?**
>
> We chose it empirically for the intuition of a higher position precision requirement for small/medium regions. Please refer to our response to ***Reviewer v6Kv Q1.*** for further discussion.
>
> **Q3. Does an imbalanced content vs position image ratio improve the results?**
>
> Thank you for your interest. We found that our method is robust to the content vs. position masking ratio. Please refer details in the table in our response to ***Reviewer v6Kv Q2.***.
>
> **To other comments**
>
> Thank you for pointing out these typos. We will fix the duplicated “Table,” add the missing reference to Figure 3, and standardize the usage of “Figure” throughout the text.

---

> > ### Comment · Reviewer_mPP8 · 2025-04-07
> >
> > Thanks for the detailed response! My follow-up comments are listed below:
> >
> > 0. I have a new concern about Table 3.b and Table 4.a. The difference between the proposed method and the comparison method is the same, but the paper includes that Table 3.b shows resolution invariance, and Table 4.a shows the advantage of cross-mask over box-mask. Only one of the two claims should hold. The marginal improvement on ViT-S with cross-wise masking undermines the necessity of using cross-wise masking, and it is also counter-intuitive that the model did not suffer a lot from the attention artifacts displayed in Figure 7. Do you have the numbers for ViT-B to justify the needs for cross-wise positional masking?
> >
> > 1. This is a great study on the potential cause of attention artifacts. The fact that simple augmentation (vertical + horizontal flipping) cannot eliminate the artifact adds value to the proposed cross-wise masking strategy.
> >
> > 2. Though classification is not the focus of this work, the IN1K result slightly weakens the proposed method's capacity, given that the proposed loss function resembles the DINOv2 loss.
> >
> > 3. The table on the task distribution ratio is clear. Having them all provides empirical ground for the 50/50 choice.

---

> > > ### Author Response · Authors · 2025-04-09
> > >
> > > Thank you for your comment regarding the comparative results in Table 3(b) and Table 4(a). Although both tables yield identical results, our interpretation appears contradictory, which we had not previously noticed.
> > >
> > > In summary, there remain measurable differences between the two methods, and higher values are preferable; hence, we continue to assert the superiority of the cross-mask. Furthermore, the reduction in attention artifacts—which, while difficult to fully capture in averages, is clearly advantageous—supports our claim.
> > >
> > > It is important to note that the aim of Table 3(b) was to demonstrate that our approach is not overly sensitive to hyperparameters, as resolution can be varied continuously. In contrast, Table 4(a) evaluates the relative merits of two masking strategies. If higher performance were observed only at a specific resolution, it would cast significant doubt on the method's overall validity.
> > >
> > > That said, we believe that higher resolution generally yields better results, which aligns with intuition. (From a computational standpoint, it is sensible to limit the resolution to a practical range—which is why we adopted a 50×50 resolution in our experiments.)
> > >
> > > In the next revision of the paper, we will clarify our interpretation of the results in Table 3(b) to ensure our intent is clearly conveyed. Although we currently do not have results for ViT-B with cross-wise masking, we will include them in the next version.

---

### Official Review · Reviewer_v6Kv · 2025-03-14

**Overall Recommendation:** 3

**Summary:**

This work extends the teacher-student SSL approach of DINO v2 with an additional task with the goal to improve the dense prediction capability of the trained model. Specifically, during training the student network is either tasked with the alignment of masked positional encoding as well as the standard alignment of masked out content views. To this end, the authors do not apply the positional with respect to just the cropped image part, but with respect to the position of the crop in the original image.

---
update after rebuttal: I think the authors should have referenced the ground work done by DropPos in a more prominent manner, as it explains the motivation behind the global mask very well. Nevertheless, the idea to sample from a virtual larger image is an interesting contribution.

**Claims And Evidence:**

The main claim of this work is that the proposed sampling of the positional encoding and the additional task of masking and predicting positional encoding vectors between a student and a teacher view improves the dense downstream performance of the model. The experiments suggest that both these additions to the DINO V2 method are beneficial.

**Essential References Not Discussed:**

None.

**Experimental Designs Or Analyses:**

Yes. Experiments and ablations are valid. Besides pretraining in a comparable setup to other methods, the finetuning in both object detection as well as segmentation downstream tasks is performed with established pipelines.

**Methods And Evaluation Criteria:**

The method is pretrained on ImageNet1k and its dense prediction performance is evaluated on COCO and ADE20K. A standard setup to compare with other SSL methods. Ablations studies focus on the object detection and instance segmentation performance on COCO.

**Other Comments Or Suggestions:**

None.

**Other Strengths And Weaknesses:**

Strengths:
- The method is simple and the authors provide a good motivation for their approach. To build upon the pipeline of DINO v2 makes a lot of sense and the authors describe the additions well. The paper is quite easy to follow.
- The improvements on the downstream evaluations on COCO over the DINO v2 baseline are significant.
- The authors provide a lot of insightful evaluations, ablation studies and qualitative examples.

Weaknesses:
- The method improves object detection on COCO, but the improvements in segmentation are limited. The fact that the position sampling on its own, even without the additional training task shows significant gains is interesting. The importance of the sampling strategy (constant, uniform, beta) is significant, but not explained or discussed extensively. Might the main benefit of the method be simply its adjustment to challenging smaller objects in the COCO dataset?

**Questions For Authors:**

Why do you use a 50/50 split between content and positional encoding prediction?  Given that training with just positional prediction does not work well and the model still needs to learn content based feature extraction with a sufficient number of training samples.

**Relation To Broader Scientific Literature:**

The focus on the role of the positional encoding itself and its utilization during pretraining are valid contributions to further improve contrastive self-supervised learning when it comes to downstream tasks that require dense prediction.

**Theoretical Claims:**

no theoretical claims.

---

> ### Author Rebuttal · Authors · 2025-04-01
>
> **Q1. The method improves object detection on COCO, but the improvements in segmentation are limited. The fact that the position sampling on its own, even without the additional training task shows significant gains is interesting. The importance of the sampling strategy (constant, uniform, beta) is significant, but not explained or discussed extensively. Might the main benefit of the method be simply its adjustment to challenging smaller objects in the COCO dataset?**
>
> As you correctly pointed out, the improvement on ADE20K is limited compared to object detection tasks. We believe this is because ADE20K is primarily a pixel-level classification task that does not require explicit spatial reasoning, and thus benefits less from relative positional encoding. (We included ADE20K to enable fair comparison with prior work.) In contrast, object detection tasks on COCO require more precise instance-level localization, which our method is explicitly designed to address.
>
> Thank you also for your valuable comment regarding the sampling strategy. To simulate the long-tail distribution of object sizes in COCO, we introduced a Beta distribution. Since small objects are frequent and more difficult to detect, we considered this distribution more realistic for the training setting. In future work, we plan to analyze the impact of different sampling distributions in a more systematic manner, and to explore their generalization to other tasks and datasets.
>
> We also appreciate your question on whether the improvement mainly stems from better handling of small objects. Indeed, we believe the position sampling is particularly effective for small objects. However, our method also improves attention diversity and instance discrimination capabilities (see Figures 5 and 6). These improvements are not limited to small objects but contribute to better object-level reasoning across a wide range of object sizes.
>
> **Q2. Why do you use a 50/50 split between content and positional encoding prediction? ...**
>
> Thank you for pointing this out. As shown in Table 4b, using position prediction alone leads to a significant drop in performance, whereas content prediction alone yields reasonably strong results. Nonetheless, we found that a 50/50 split between content and position masking achieves the best overall performance (see the table below for more details).
>
> We hypothesize that the effectiveness of this balanced setup stems from the fact that the position prediction task, while insufficient on its own, encourages the model to explicitly encode spatial relationships, which synergizes with content learning. The 50% allocation provides a sufficient signal for content feature extraction while still allowing position prediction to serve as a meaningful auxiliary task. We agree that this trade-off warrants further exploration, and we plan to investigate more dynamic or adaptive scheduling strategies in future work.
>
> | Content vs. Pos. | 100/0 | 75/25 | 50/50 | 25/75 | 0/100 |
> |------------------|-------|-------|-------|-------|--------|
> | **AP$^\text{box}$**  | 43.4  | 44.2  | 44.8  | 44.2  | 21.7   |
> | **AP$^\text{mask}$** | 38.9  | 39.5  | 39.8  | 39.5  | 20.7   |

---

### Decision · Program_Chairs · 2025-05-01

**Decision:**

Accept (poster)

**Comment:**

The submission discusses a method for self-supervised feature learning for object detection. Initially, the reviewers raised concerns about the similarity to DropPos, limited gains on segmentation, somewhat limited experimentation, limited novelty (DenseCL, LOCA, DropPos, etc.), and limited comparison to MAE-style methods. After the rebuttal and a lengthy discussion the reviewers arrived at an accept recommendation. The reviewers recognized the merits of the work while pointing out that the rebuttal wasn't able to resolve all of the aforementioned concerns (e.g., limited comparison to MAE-style methods). AC concurs with an unanimous accept recommendation and recommends a weak accept.